

# Genome-wide analysis of the strigolactone biosynthetic and signaling genes in grapevine and their response to salt and drought stresses

Yanyan Yu[1,*], Jinghao Xu[1,*], Chuanyin Wang[2], Yunning Pang[1], Lijian Li[1], Xinjie Tang[1], Bo Li[3] and Qinghua Sun[1]

[1] College of Life Science, State Key Laboratory of Crop Biology, Shandong Agricultural University, Taian, Shandong, China
[2] Forestry Bureau of Heze, Heze, Shandong, China
[3] Shandong Academy of Grape, Shandong Academy of Agricultural Sciences, Jinan, Shandong, China
* These authors contributed equally to this work.

## ABSTRACT

Strigolactones (SLs) are a novel class of plant hormones that play critical roles in regulating various developmental processes and stress tolerance. Although the SL biosynthetic and signaling genes were already determined in some plants such as *Arabidopsis* and rice, the information of SL-related genes in grapevine (*Vitis vinifera* L.) remains largely unknown. In this study, the SL-related genes were identified from the whole grapevine genome, and their expression patterns under salt and drought stresses were determined. The results indicated that the five genes that involved in the SL biosynthesis included one each of the *D27*, *CCD7*, *CCD8*, *MAX1* and *LBO* genes, as well as the three genes that involved in the SL signaling included one each of the *D14*, *MAX2*, *D53* genes. Phylogenetic analysis suggested that these SL-related proteins are highly conserved among different plant species. Promoter analysis showed that the prevalence of a variety of *cis*-acting elements associated with hormones and abiotic stress existed in the promoter regions of these SL-related genes. Furthermore, the transcription expression analysis demonstrated that most SL-related genes are involved in the salt and drought stresses response in grapevine. These findings provided valuable information for further investigation and functional analysis of SL biosynthetic and signaling genes in response to salt and drought stresses in grapevine.

# INTRODUCTION

Salt and drought stresses are the major abiotic stress factors that seriously affect the plant growth and development worldwide (*Qin et al., 2017*). Due to their sessile lifestyle, plants have developed a series of changes in metabolism, physiology and biochemical mechanisms to adapt to the osmotic stress. Previous studies have revealed that some plant hormones, such as auxin (IAA) (*Naser & Shani, 2016*; *Bielach, Hrtyan & Tognetti, 2017*),

Corresponding authors
Bo Li, sdtalibo@163.com
Qinghua Sun, qhsun@sdau.edu.cn

brassinolide (BR) (*Hu et al., 2016*; *Talaat & Shawky, 2016*), gibberellin (GA) (*Colebrook et al., 2014*), abscisic acid (ABA) (*Fricke et al., 2004*; *Osakabe et al., 2014*) and strigolactones (SLs) (*Min et al., 2019*; *Ling et al., 2020*; *Zulfiqar et al., 2021*), play important roles in response to abiotic stress (*Cheng, Ruyter-Spira & Bouwmeester, 2013*).

SLs, as new type of plant hormones, were originally identified as a root secretory compound promoting the seed germination of parasitic weeds, such as *Striga lutea* (*Cook et al., 1966*). Subsequently, SLs have been proved to play a pivotal role in regulating root architecture, repressing shoot branching and enhancing leaf senescence (*Umehara et al., 2008*). Recently, more and more researchers demonstrated that SLs could positively regulate plant response to abiotic stress, such as salinity (*Zheng et al., 2021*), drought (*Min et al., 2019*) and chilling stresses (*Cooper et al., 2018*). In the past decade, the functions of SLs have been extensively studied, and some genes involved in the SLs biosynthetic and signal transduction pathways have been identified (*Wu et al., 2019*; *Qiao et al., 2020*).

SLs are biosynthesized from carotenoid pathways involving at least five enzymes, including a cis/trans-carotene isomerase DWARF27 (D27), two carotenoid cleavage dioxygenases (CCD7 and CCD8), a cytochrome P450 monooxygenase MORE AXILLARY GROWTH 1 (MAX1) and an oxidoreductase-like enzyme LATERAL BRANCHING OXIDOREDUCTASE (LBO) (*Lopez-Obando et al., 2015*; *Brewer et al., 2016*). Initially, D27 catalyzed the reversible isomerization of trans-β-carotene into 9-cis-β-carotene. Next, the 9-cis-β-carotene was converted into 9-cis-β-apo-10′-carotenol, which is subsequently catalyzed into the SLs precursor carlactone (CL) by the CCD8. Among different plant species, the biosynthesis pathway of CL from trans-β-carotene is the same, while the biosynthesis process from CL to active SLs may varies (Fig. 1). For example, in *Arabidopsis*, CL is catalyzed by MAX1 to produce carlactonoic acid (CLA), which is subsequently methylated to form methyl carlactonoate (MeCLA) by a methyl transferase; then, MeCLA was converted into a bioactive SL by LBO. However, in rice, CL is catalyzed by the MAX1 homolog Os900 (CYP711A2) to generate CLA, which is further converted into 4-deoxyorobanchol (4DO) by CYP711A2. Subsequently, 4DO was catalyzed by another MAX1 homolog (CYP711A3) to produce orobanchol that has SL activity (*Wu et al., 2019*).

In recent years, mounting studies have provided essential insights into the SL signal transduction pathway, and several major components in this pathway were identified, including dwarf 14 (D14), Dwarf 3 (D3)/more axillary growth (MAX2), Dwarf 53 (D53)/ SMXLs, TOPLESS (TPL)-related protein two (TPR2), and Ideal Plant Architecture 1 (IPA1) (*Waters et al., 2017*; *Yoneyama & Brewer, 2021*). D14 encodes a α/β-hydrolase fold protein, which acts as an SL receptor to bind and hydrolyze SL (*Yao et al., 2016*). MAX2 is an F-box leucine-rich repeat (LRR) protein, which acts as the substrate-recruiting subunit of the SCF (SKP, Cullin, and F-box protein) complex (*Stirnberg, Furner & Ottoline Leyser, 2007*). D53 encodes a class I Clp ATPase proteins, which has been identified as a SL signaling repressor (*Zhou et al., 2013*). In the presence of SLs, D14 could interact with MAX2 to form a D14-SCF ubiquitination complex, resulting in the degradation of the nuclear-localized repressor D53 and activates the repressed downstream SL signaling. Conversely, in the absence of SLs, D14 cannot interact with MAX2 and D53, while D53 can

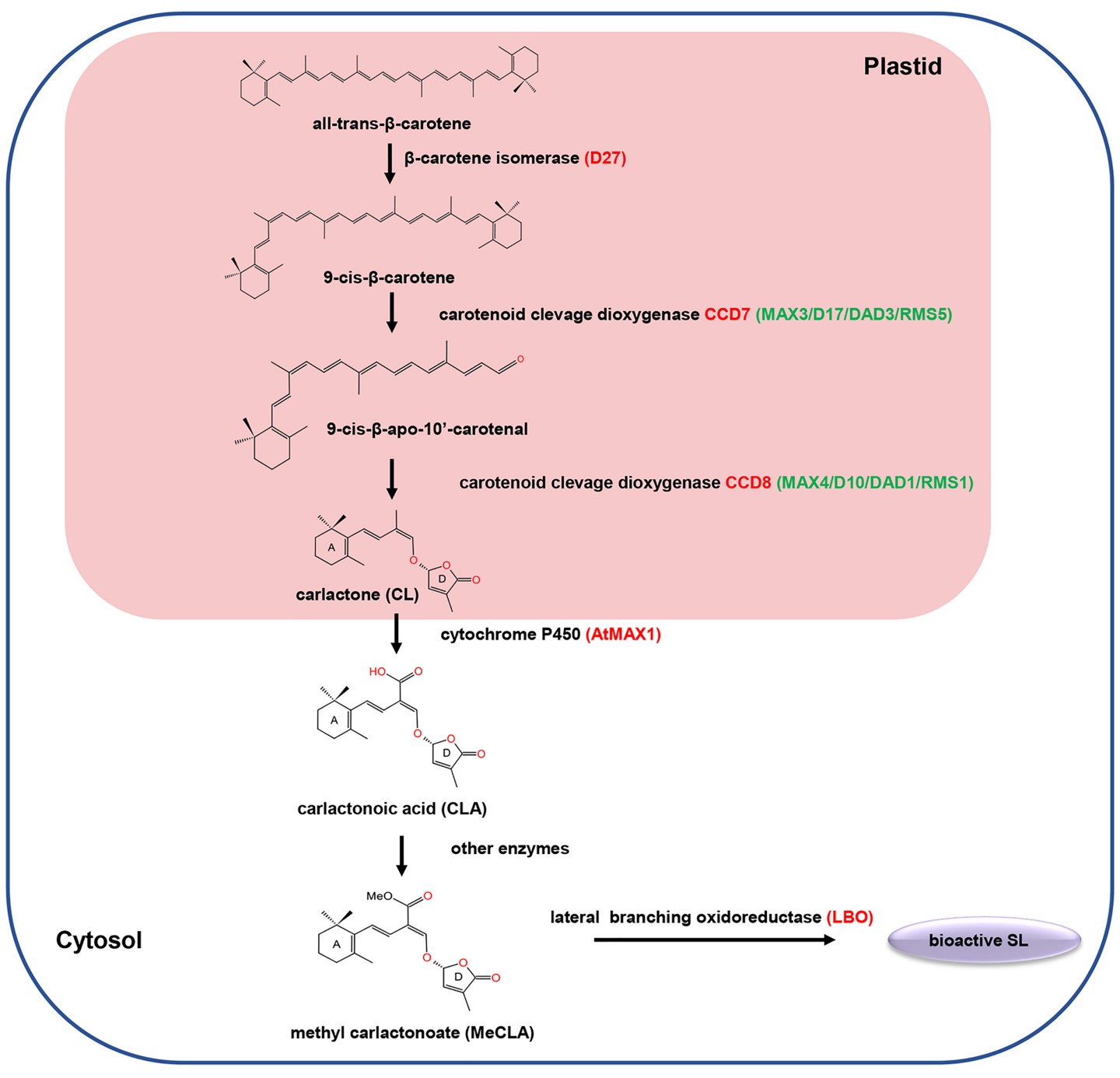

**Figure 1 The carotenoid pathway diagram.**

form the complex with TPR2 and IPA1 to repress the expression of IPA1, subsequently suppressing the expression of IPA1-regulated genes and no SL response (*Jiang et al., 2013*; *Song et al., 2017*).

Recent studies have revealed that some SL biosynthetic and signaling genes play critical roles in abiotic stresses response. For instance, the *Arabidopsis max* mutants, including *max2*, *max* 3 (CCD7) and *max* 4 (CCD8), displayed hypersensitivity to salt and drought

stresses than wild type (WT) plants (*Ha et al., 2014*). Overexpression of apple *MdMAX2* in *Arabidopsis* improved plant tolerance to salt and drought stress (*An et al., 2016*). Overexpression *Sapium sebiferum SsMAX2* in *Arabidopsis* also significantly promoted the plant resistance to drought and salt stresses (*Wang et al., 2019*). In soybean, the expression of *GmCCD7b, GmMAX1c, GmMAX2a/b* and *GmD53a/b* were all significantly upregulated under salt stress (*Qiao et al., 2020*). In addition, *D14* mutant *Arabidopsis* were more susceptible to drought than WT plants, while overexpression apple *MdD14* in *Arabidopsis* significantly enhanced the plant tolerance to salt and drought stresses (*Yang et al., 2019*). All the above studies indicated that SL-related genes are closely linked to plant stress resistance.

Grapevine is one of the most economically important fruit crops, which is widely cultivated in the world. However, some extreme environmental conditions, such as salinity and drought, adversely affect the grapevine growth and productivity (*Cramer et al., 2007*). Hence, discovering novel genes involved in abiotic stress resistance and application of genetic breeding are considered as an effective way to improve grapevine resistance. The existence of a high-quality *de novo* assembled grapevine genome has made it possible to identify gene families in this species (*Jaillon et al., 2007*). Although previous studies have identified *CCD7* and *CCD8* genes in grapevine and revealed their role in grapevine shoot branching (*Ren et al., 2020*). However, the whole-grapevine SL biosynthetic and signaling genes have not been identified so far. Moreover, their functions in response to salt and drought stresses remain largely unclear. Therefore, in the present study, we performed a genome-wide search of SL-related genes in grapevine genome and analyzed their phylogenetic relationships, gene structures, chromosomal distributions, and promoter *cis*-elements. Subsequently, their expression patterns in different organs were checked. Furthermore, we examined their expression patterns in response to salt and drought stresses. The results provide a valuable information for future functional analyses of SL biosynthetic and signaling genes.

## MATERIALS AND METHODS

### Identification of SL biosynthetic and signaling genes in grapevine

To identify the SL biosynthetic and signaling genes in grapevine, genes and proteins annotated in *A. thaliana*, rice, soybean, peach and apple were obtained from Phytozom (https://phytozome.jgi.doe.gov/pz/portal.html). The grapevine reference genome assembly (12X.v2), including protein sequence database, CDS database, genome sequence database, and v2. annotation file, were downloaded from the Grapevine Genome Browser (12×) (http://www.genoscope.cns.fr/externe/GenomeBrowser/Vitis/). The protein sequences of SL biosynthetic and signaling genes of *Arabidopsis* and rice were used as queries for identified candidate SL-related genes of grapevine in grapevine genome *via* basic local alignment search tool (BLAST). Then, all the potential candidates of SL-related genes obtained were submitted to SMART (http://smart.embl-heidelberg.de/) and HMMER (https://www.ebi.ac.uk/Tools/hmmer/search/hmmscan) online software to identify the conserved domains. The incomplete and redundant sequences were removed. Further, the isoelectric point (pI) and molecular weight (MW) of these SL biosynthetic and signaling

proteins were predicted by the ExPASy (https://web.expasy.org/protparam/) online software. The subcellular localization of grapevine SL biosynthetic and signaling genes was predicted with Cell-PLoc 2.0 (http://www.csbio.sjtu.edu.cn/bioinf/Cell-PLoc-2/) software.

## Protein conserved motifs and phylogenetic relationships analysis of the SL genes

The conserved motifs of grapevine SL biosynthesis and signal protein were identified at Multiple EM for Motif Elicitation (MEME) (https://meme-suite.org/meme/) online website by comparison with *Arabidopsis*, rice, soybean, peach, or apple SL related proteins, respectively. Full-length sequences of SL-related proteins of grapevine, *Arabidopsis*, rice, soybean, peach and apple were aligned using the Clustal W program within MEGA 6.0 software. The phylogenetic tree was constructed by the neighbor-joining (NJ) method with 1,000 bootstrap replications and p-distance model and validated by the Maximum likehood method. To better visualize the phylogenetic tree, the final tree diagram file (*.nwk) was uploaded from MEGA to the Figtree v14.3 and iTOL (https://itol.embl.de/) online software.

## Chromosomal localization and synteny analysis of grapevine SL related genes

The chromosomal locations data of grapevine SL-related genes were retrieved from the Grapevine Genome Browser (12×) (http://www.genoscope.cns.fr/externe/GenomeBrowser/Vitis/). The chromosome map showing the physical location of grapevine SL-related genes was drawn using TBtools software (v1.082) (*Chen et al., 2020*). To explore the synteny relationship of the orthologous SL-related genes of grapevine with *Arabidopsis* and rice, the genome sequence files and annotation files of *Arabidopsis* and rice were obtained from Phytozome database (v12, https://phytozome.jgi.doe.gov/pz/portal.html). Then, the Multiple Collinearity Scan toolkit (MCScanX) (https://github.com/tanghaibao/jcvi/wiki/MCscan-(Python-version)) was used to analyze the gene replication events with default parameters. The synteny maps were generated using TBtools software (v1.082).

## *Cis*-acting elements analysis in the grapevine SL related genes promoter sequences

The DNA sequence within 1,500-bp upstream of the initiation codon (ATG) of all SL biosynthesis and signal transmission genes in grapevine was obtained from the Ensemble plants database (http://plants.ensembl.org/index.html). Then, these sequences were uploaded to the PlantCARE Web Tools (http://bioinformatics.psb.ugent.be/webtools/plantcare/html/) as promoter sequences to detected the *cis*-acting elements (*Lescot et al., 2002*).

## Organ-specific expression analysis of grapevine SL related genes

To explore the tissue-specific expression patterns of grapevine SL biosynthesis and signal transmission genes, the organ expression profile of grapevine (GSE36128) was obtained

from the Gene Expression Omnibus (GEO) database (https://www.ncbi.nlm.nih.gov/geo/). According to all the SL related genes ID, their expression data were extracted from the GSE36128 datasets. The heatmap and clustering tree were constructed and visualized with software TBtools (v1.082).

## Plant growth and stress treatments

The tissue culture seedlings of grapevine 'Crimson seedless' was cultured in Murashigeand Skoog (MS) solid medium with 0.2 mM indole-3-butytric acid (IBA) under a 16 h light/8 h dark cycle at 24 °C for 4-week intervals. Then, the uniformly grapevine shoot seedlings were selected and transferred to the liquid medium containing 200 mM NaCl or 200 mM mannitol for salt and drought stress treatments. The treated seedlings were sampled at 0, 3, 6 and 12 h, and immediately frozen in liquid nitrogen and stored in at −80 °C for RNA extraction. Each treatment was repeated at least three times.

## RNA extraction and quantitative real-time PCR

The total RNA was extracted from samples treated with NaCl using HiPure HP Plant RNA Mini Kit (Magen, shanghai, China). Then, the RNA was used for cDNA synthesis with the PrimeScript™ RT reagent kit with gDNA Eraser (Vazyme, Nanjing, China). Quantitative real-time PCR (qRT-PCR) was performed according to the supplier's instructions of the SYBR® Green Premix Pro Taq HS qPCR Kit (Accurate Biotechnology, Hunan, China) in the CFX96TM Real-Time PCR Detection System (Bio-Rad, Hercules, CA, USA). The PCR reaction system consists of one 5 μL cDNA sample, 0.6 μL primers, and 7.5 μL SYBR Green (Accurate Biotechnology, Hunan, China), 1.9 μL nuclease-free $H_2O$, and the reaction volume was 15 μL. The PCR reaction was performed with the following conditions: 30 s at 95 °C, 40 cycles of 5 s at 95 °C, and 30 s at 60 °C. The grapevine β-actin (XM_034827164.1) were used as the internal references. All experiments were repeated at least three times and all the primers used in this study were listed in Table S1.

## Statistical analysis

The experiment data were automatically analyzed by the CFX Manager software program using the $2^{-\Delta\Delta CT}$ comparative CT method (*Livak & Schmittgen, 2001*). Statistical significance was analyzed using Duncan's multiple range tests with analysis of variance (ANOVA), and calculations were performed with SPSS Statistics 19.0. Significance was set at $p < 0.05$.

# RESULTS

## Identification of SL biosynthetic gene D27, CCD7, CCD8, MAX1 and LBO in grapevine

Given SL biosynthetic genes have already been identified in some plants, such as *Arabidopsis*, rice, and soybean (*Waters et al., 2017*; *Qiao et al., 2020*). These protein sequences of *Arabidopsis* and rice were used as queries to retrieve the similar proteins in grapevine genome databases. By removing incomplete and redundant sequences, one each

**Table 1 Physicochemical properties of grapevine SL biosynthetic and signaling genes.**

| Name | Gene ID | CDS length (bp) | Protein (aa) | MW (kDa) | pI | GRAVY | Subcellular localization |
|------|---------|-----------------|--------------|----------|-----|-------|--------------------------|
| SL biosynthetic gene | | | | | | | |
| VvD27 | VIT_200s0179g00330.1 | 840 | 279 | 31.48785 | 9.12 | −0.086 | Chloroplast. Nucleus |
| VvCCD7 | VIT_215s0021g02190.1 | 1,833 | 610 | 68.46228 | 7.31 | −0.313 | Cytoplasm |
| VvCCD8 | VIT_204s0008g03380.1 | 1,692 | 563 | 62.34390 | 7.27 | −0.339 | Chloroplast |
| VvMAX1 | VIT_204s0008g01100.1 | 1,605 | 534 | 60.28409 | 9.27 | −0.151 | Endoplasmic reticulum |
| VvLBO | VIT_205s0020g01310.1 | 1,098 | 365 | 41.46443 | 5.56 | −0.320 | Cytoplasm |
| SL signaling gene | | | | | | | |
| VvMAX2 | VIT_212s0028g02140.1 | 2,139 | 712 | 78.80740 | 6.01 | −0.106 | Nucleus |
| VvD14 | VIT_218s0001g09140.1 | 801 | 266 | 29.45509 | 6.30 | 0.250 | Cell membrane. Chloroplast |
| VvD53 | VIT_206s0004g06700.1 | 3,321 | 1,106 | 120.86401 | 6.66 | −0.285 | Chloroplast |

of *D27*, *CCD7*, *CCD8*, *MAX1* and *LBO* genes were obtained from grapevine genome. The detailed information regarding these identified SL biosynthetic proteins was provided in Table 1, including the gene name, gene ID, length of coding DNA sequence (CDS), length of protein sequence, molecular weight (MW), theoretical isoelectric point (pI) and subcellular localization.

To evaluate the evolutionary relationships of grapevine SL biosynthetic proteins with these proteins in other plants, the phylogenetic trees were constructed using the protein sequences of these genes with *Arabidopsis*, rice, soybean, peach, or apple. The results showed that VvD27, VvCCD8 and VvMAX1 shared higher homology with the same proteins in soybean, respectively. Notably, the tissue-specific expression patterns of soybean homologs of VvD27, VvCCD8, and VvMAX1 were similar to those in grapevine (Fig. S1), which further supported the above result. In addition, VvCCD7 was close to *Arabidopsis* (AtCCD7), and VvLBO shared higher homology with peach protein (Prupe.8G233300) (Fig. 2). Further, multiple sequence alignment analysis revealed that all D27 (VvD27, GmD27a/b, AtD27, and OsD27), CCD7 (VvCCD7, GmCCD7a/b, AtCCD7, and OsCCD7), CCD8 (VvCCD8, GmCCD8a/b, AtCCD8, and OsCCD8) and MAX1 (VvMAX1, GmMAX1a/b/c, AtMAX1, and OsMAX1a/b/c) proteins exhibit the same motif composition, respectively. Similarly, LBO proteins of grapevine (VvLBO), *Arabidopsis* (AtLBO), rice (OsLBO), apple (MDP0000211165 and MDP0000152548), peach (1G138800) and maize (GRMZM2G025870) also displayed nearly the same motifs (Fig. 2). These findings suggest that these SL biosynthetic genes are highly conserved among different plant species.

## Identification of SL signaling genes D14, MAX2 and D53 in grapevine

To identify SL signaling genes in grapevine, the D14, MAX2, or D53 protein sequences of *Arabidopsis* and rice were used as queries to retrieve the similar proteins in grapevine genome databases. After removing incomplete and redundant genes, one each of *D14*, *MAX2* and *D53* were identified in grapevine. The detailed information of these SL signaling genes was presented in Table 1. Phylogenetic tree analysis revealed that VvMAX2

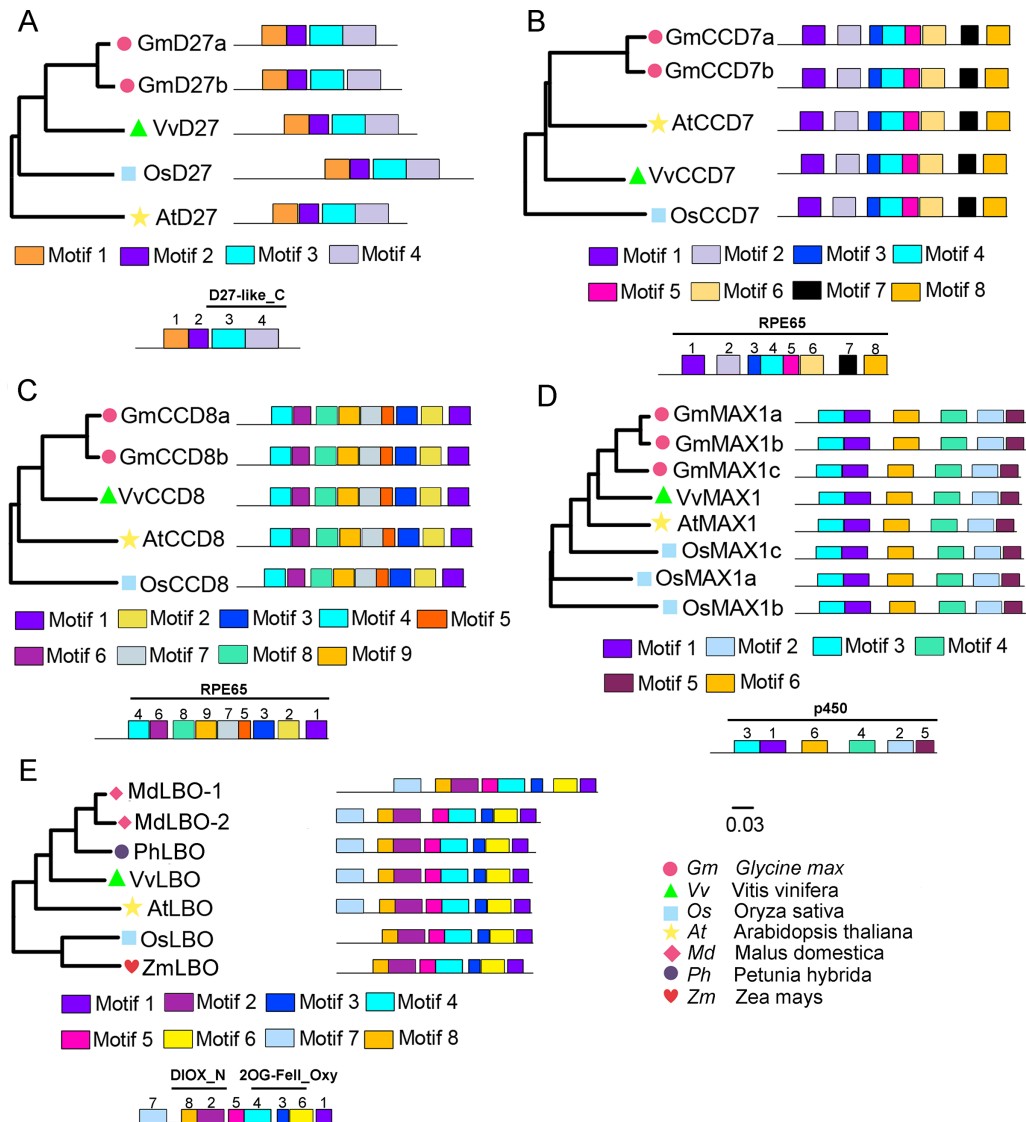

**Figure 2 Phylogenetic relationships and conserved motifs analysis of SL biosynthetic genes from different plant species.** (A) D27, (B) CCD7, (C) CCD8, (D) MAX1, (E) LBO. The conserved motif sequences were predicted at MEME online website and indicated by different colored squares. The protein sequences for conserved motifs are shown in Figs. S2 & S3. The phylogenetic trees were constructed using the neighbor-joining (NJ) method on MEGA 6.0 software. Gm, *Glycine max*; Vv, *Vitis vinifera*; At, *Arabidopsis thaliana*; Os, *Oryza sativa*; Md, *Malus domestica*; Ph, *Petunia hybrida*; Zm, *Zea mays*.

and VvD53 shared higher homology with *Arabidopsis* (AtMAX2) and soybean (GmD53c), respectively. In addition, MEME online analysis indicated that all MAX2 (VvMAX2, GmMAX2a/b, AtMAX2, and OsMAX2) and D53 (VvD53, GmD53a/b/c, AtD53, and OsD53a/b) displayed the same motif composition, respectively (Fig. 3).

D14 encodes an α/β-hydrolase fold protein, which acts as an SL receptor in the plant SL signaling pathway, while D14L (KAI2/HTL/D14-Like) is the receptor protein in plant KAR (karrikins) signaling pathway, which is a close homologue of D14 (*Qiao et al., 2020*).

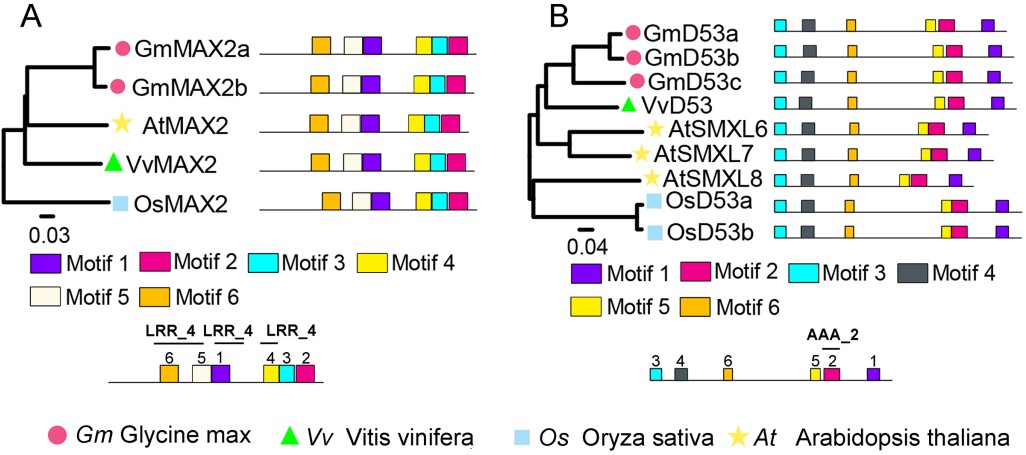

**Figure 3** **Phylogenetic relationships and conserved motifs analysis of MAX2 and D53 families in** *Glycine max, Vitis vinifera, Arabidopsis thaliana* **and** *Oryza sativa*. (A) The phylogenetic and motif analysis of MAX2. (B) The phylogenetic and motif analysis of D53. Six conserved motifs were identified using MEME and displayed by different colored boxes. The protein sequences for conserved motifs are shown in Fig. S4.

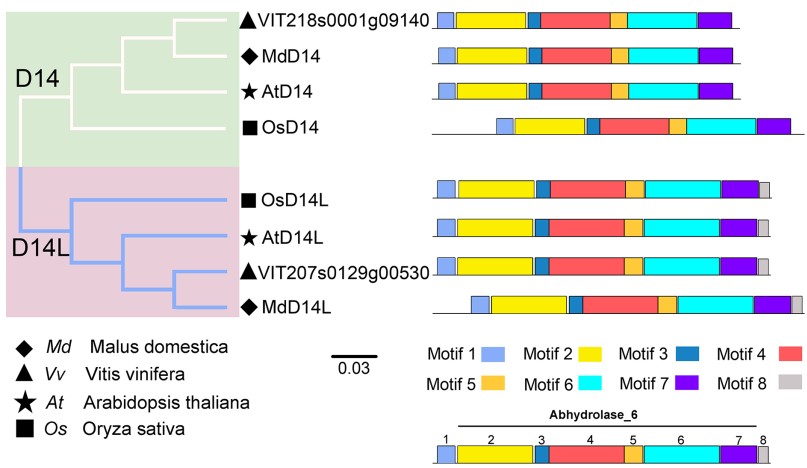

**Figure 4** **The phylogenetic tree and conserved motifs analysis of D14 and D14L protein family.** The conserved sequences were displayed using eight colored boxes. The protein sequences for conserved motifs are shown in Fig. S5. (apple) *Malus domestica*, (grapevine) *Vitis vinifera*, (*Arabidopsis*) *Arabidopsis thaliana*, (rice) *Oryza sativa*.

In this study, the D14 proteins from rice (OsD14) and *Arabidopsis* (AtD14) were used as queries to identify homologs in grapevine genome databases. By removing incomplete and redundant sequences, two complete domain containing proteins were identified from grapevine protein database. To further identify D14 proteins in grapevine, a phylogenetic tree was constructed using the D14 and D14L proteins in rice, *Arabidopsis*, and apple. The results showed that these proteins were classified into two clades, D14 and D14L. One grapevine protein (VIT_207s0129g00530) was clustered in clade D14L with the KAR receptors AtD14L, OsD14L and apple D14L protein (MDP0000136111), while another grapevine gene (VIT_218s0001g09140) was clustered in clade D14 with the SL receptors AtD14, OsD14 and apple D14 protein (MDP0000529739) (Fig. 4). Therefore,

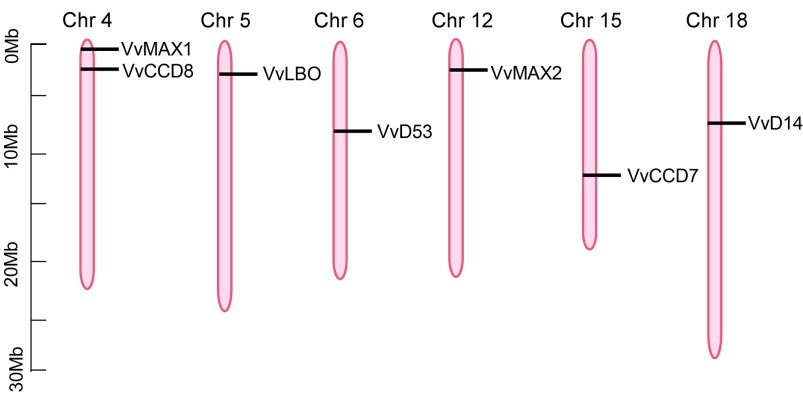

**Figure 5 The positioning of SL biosynthetic and signaling genes on chromosomes.** The chromosome name is located at the top of each bar. The length of chromosomes is displayed in millions of bases (Mb).

VIT_218s0001g09140 gene was named *VvD14*, and VIT_207s0129g00530 gene was named *VvD14L*. To further investigate the difference between D14 and D14L during evolution, MEME online software was used to analysis their motif composition. As shown in Fig. 4, all D14 and D14L proteins appeared with the similar motif composition, which all comprise with motif 1–7; however, only D14L proteins contain motif 8, suggesting that the functions of D14 and D14L proteins may be different.

## Chromosomal locations of SL related genes in grapevine

To investigate the chromosomal locations of SL biosynthetic and signaling genes, the annotation information was obtained from the Grapevine Genome Browser database and determined by TBtools software. As shown in Fig. 5, seven putative SL related genes were randomly distributed on six chromosomes, including chromosome 4, 5, 6, 12, 15, 18, however, *VvD27* was failed to be anchored to any particular chromosomes, which could be localized to the scaffolds of an unassembled genomic sequences. *VvCCD8* and *VvMAX1* were present on chromosome four, *VvLBO* was located on chromosome five, *VvD53* was anchored to chromosome six, *VvMAX2* was present on chromosome 12, *VvCCD7* on chromosome 15, *VvD14* was localized on chromosome 18 (Fig. 5).

## Synteny analysis of grapevine SL related genes with *Arabidopsis* and rice

To further analysis the phylogenetic mechanisms of grapevine SL biosynthetic and signaling genes with *Arabidopsis* and rice, two comparative syntenic maps were constructed (Fig. 6). A total of four grapevine SL related genes displayed collinear relationship with six SL related genes in *Arabidopsis*, while four grapevine SL related genes exhibited collinear relationship with five SL related genes in rice. For *Arabidopsis*, *VvCCD7*, *VvCCD8 and VvMAX2* associated with *AtCCD7*, *AtCCD8* and *AtMAX2*, respectively. Only *VvD53* were associated with three SL related genes, including *AtSMXL6*, *AtSMXL7* and *AtSMXL8*. Similarly, for the collinear gene pairs between grapevine and rice,

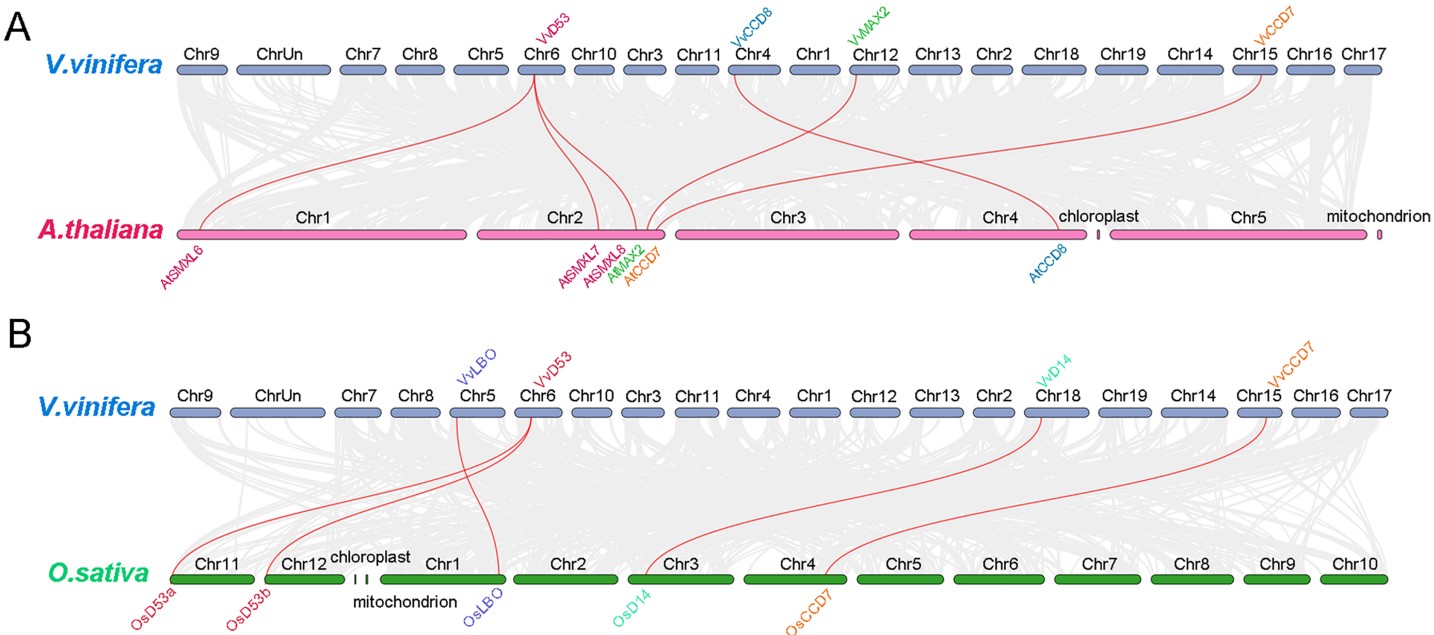

**Figure 6 Synteny analysis of grapevine SL related genes with *Arabidopsis* and rice.** (A) Synteny analysis of SL-related genes between grapevine and *Arabidopsis*, (B) synteny analysis of grapevine and rice SL-related genes. Gray lines represent the collinear blocks between two species and red lines represent the syntenic gene pairs between two species. The collinearity findings are listed in Tables S3 and S4.

three SL related genes (*VvCCD8*, *VvLBO* and *VvD14*) are associated with only one SL related genes of rice (OsCCD8, OsLBO and OsD14, respectively), while *VvD53* showed a collinear relationship with two rice SL related genes, *OsD53a* and *OsD53b*. Notably, *VvD53* and *VvCCD7* are colinear in both *Arabidopsis* and rice (Fig. 6).

## Analysis of *cis*-acting elements of SL biosynthetic and signaling gene promoters in grapevine

To investigate the potential roles of grapevine SL related genes under abiotic stresses, the 1,500 bp upstream sequences of these genes were uploaded to the PlantCARE online software (http://bioinformatics.psb.ugent.be/webtools/plantcare/html/). The functions of all *cis*-acting elements are listed in Table S2. The results revealed that many putative *cis*-acting elements related to abiotic stress-response were abundant in the promoter region of these genes, including ERE (Ethylene responsive element), ARE (anaerobic responsive elements), MYB (MYB responsive element), MYC (MYC responsive element) and TC-rich repeats. All of the SL related genes contained plant hormone-responsive elements, such as SA -responsive element (TCA), ABA-responsive (ABRE) and MeJA response element (TGACG-motif and CGTCA-motif). In addition, several *cis*-elements involved in low temperature (LTR) and light responsiveness (MRE) also were observed in the promoters of these SL related genes (Fig. 7). This result demonstrates that these SL related genes may play important roles in plant development and stresses response.

**Figure 7 The distribution of *cis*-acting elements in the promoter sequences of grapevine SL-related genes.** The different colors and circle sizes show the number of *cis*-acting elements of SL-related genes. The deeper red circle indicates the greater the number of *cis*-acting elements, the lighter red circle indicates the smaller the number.

## Organ-specific expression pattern of SL biosynthetic and signaling genes in grapevine

To examine the organ-specific expression of grapevine SL biosynthetic and signaling genes, the expression data of these genes were extracted from the expression profile of grapevine (GSE36128). As shown in Fig. 8, the expression of these SL related genes was different in different organs. *VvMAX1* showed higher expression in roots, berries, seeds and buds. *VvD27* and *VvCCD7* mainly expressed in berry and buds. The expression of *VvLBO* was most highly in roots, stems, leaves, flowers and seed. However, *VvCCD7* was expressed at significantly lower levels in these identified tissues. Moreover, *VvMAX2* and *VvD14* were higher expressed in all the tissues except berries, and *VvD53* was only highly expressed in leaves and flowers. Taken together, these SL biosynthetic and signaling genes were ubiquitous in the tissues of grapevine, which may participate in the plant growth and development.

## Expression profiles of SL biosynthetic and signaling genes in response to salt and drought stresses

Mounting evidence suggests that SL biosynthetic and signaling genes play an important role in plant responses to abiotic stress (*An et al., 2016*; *Cheng et al., 2018*; *Wang et al., 2019*). To further explore the potential functions of SL related genes in grapevine under salt (NaCl) and drought (mannitol) stresses, the expression profiles of grapevine SL biosynthetic and signaling genes were measured using qRT-PCR. The results indicated that almost all the SL biosynthetic and signaling genes responded to salt and drought stresses. Under NaCl treatment, eight genes (*VvD27*, *VvCCD7*, *VvCCD8*, *VvMAX1*, *VvD14*, *VvMAX2*, *VvD53* and *VvLBO*) were upregulated both in grapevine leaves (Fig. 9) and roots (Fig. 10). Among them, *VvCCD7*, *VvMAX2* and *VvD53* showed a significantly higher expression level at 6 h both in grapevine leaves and roots under salt stress. Meanwhile, two (*VvD27* and *VvMAX1*) SL biosynthetic genes displayed obviously higher expression at 6 h in roots and at 12 h in leaves. In addition, *VvCCD8*, *VvD14* and *VvLBO* showed a significantly higher expression level at 3 h in roots and 6 h in leaves (Figs. 9 and 10). Under mannitol treatment, six genes (*VvD27*, *VvCCD7*, *VvCCD8*, *VvMAX1*, *VvMAX2* and *VvD53*) showed a significantly increased expression both in grapevine leaves and roots (Figs. 11 and 12), whereas the expression of *VvLBO* and *VvD14* was showed a decrease in roots. What's more, *VvCCD7*, *VvCCD8* and *VvMAX2* all showed a significantly higher expression level at 6 h both in leaves and roots, and the maximum expression level of *VvD27* were observed at 3 h in roots and 6 h in leaves under drought stresses (Figs. 11 and 12). In conclusion, these results suggested that these SL biosynthetic and signaling genes of grapevine may play an important role in the salt and drought stresses response.

## DISCUSSION

Salt and drought stresses are major abiotic limiters affecting plant development worldwide. Mounting studies reveals that SLs, a new class of plant hormones, play critical roles in plant development and abiotic stress responses (*Cooper et al., 2018*; *Min et al., 2019*; *Zheng et al., 2021*), and the SL biosynthetic and signaling transduction genes have been identified

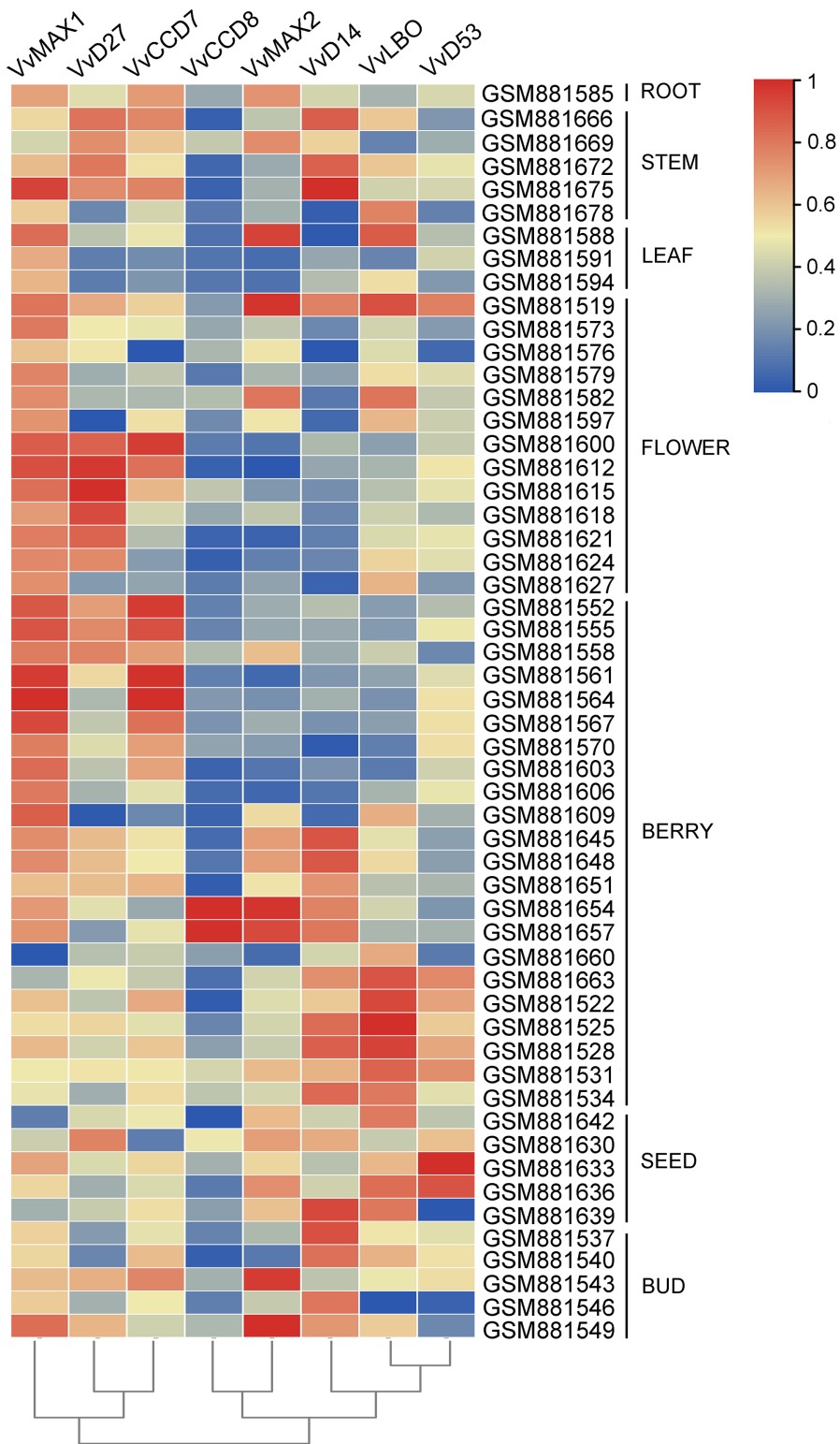

**Figure 8 The expression pattern analysis of SL biosynthetic and signaling genes in different organs of grapevine.** The expression data of SL biosynthetic and signaling genes was downloaded from Gene Expression Omnibus (GEO) database with GEO accession number GSE36128. The heatmap was drawn using TBtools software (v1.082) where the red squares represent higher expression and blue squares represents lower expression.

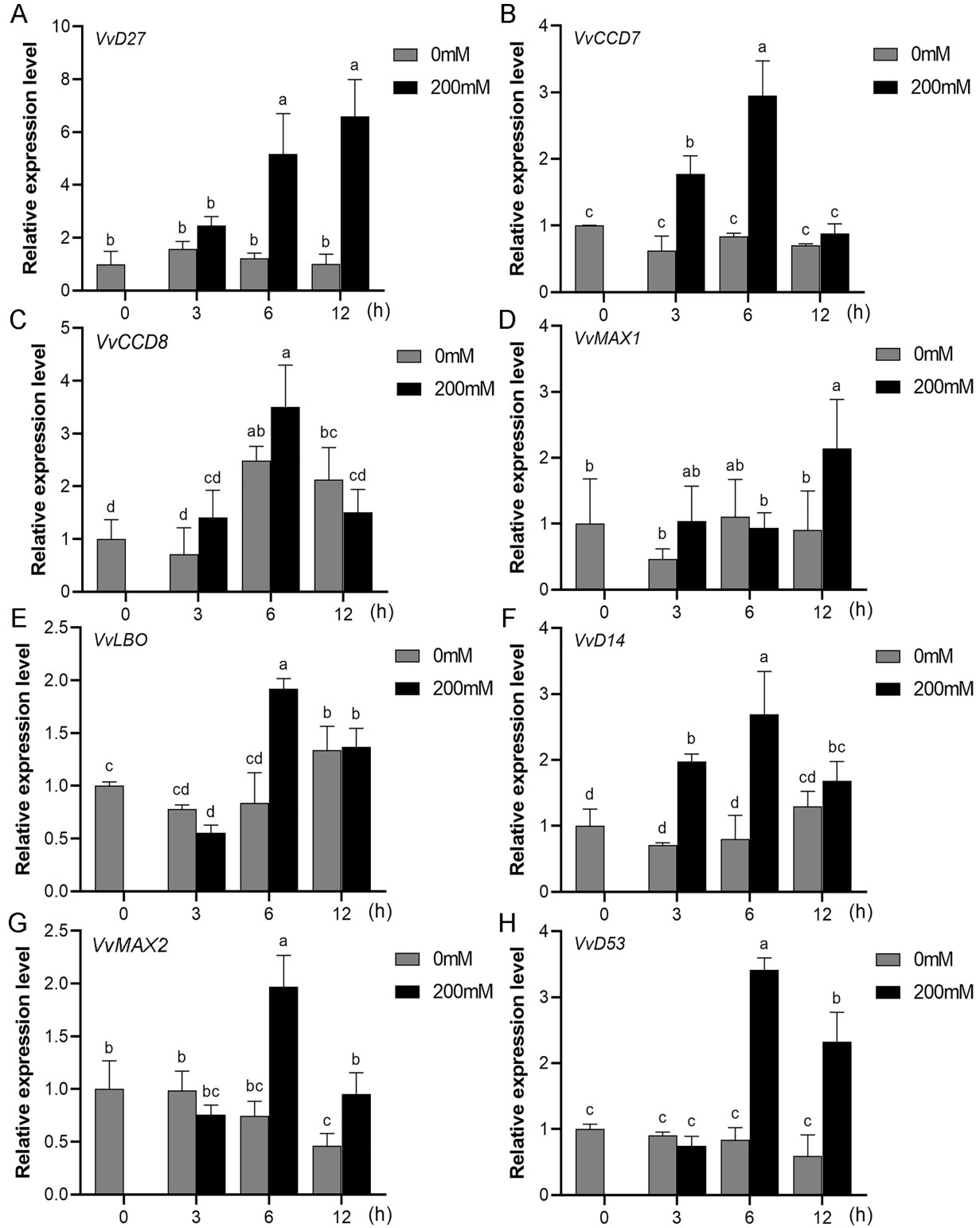

**Figure 9 The expression analysis of grapevine SL biosynthetic and signaling genes in leaves under salt stress.** (A) *VvD27*, (B) *VvCCD7*, (C) *VvCCD8*, (D) *VvMAX1*, (E) *VvLBO*, (F) *VvD14*, (G) *VvMAX2*, (H) *VvD53*. Grape seedlings were treated with 200 mM NaCl for 0, 3, 6 and 12 h, the leaves of control and treated grapevine seedlings were selected for RT-qPCR analysis for the same treatment time. The mean expression value was calculated from three independent replicates.

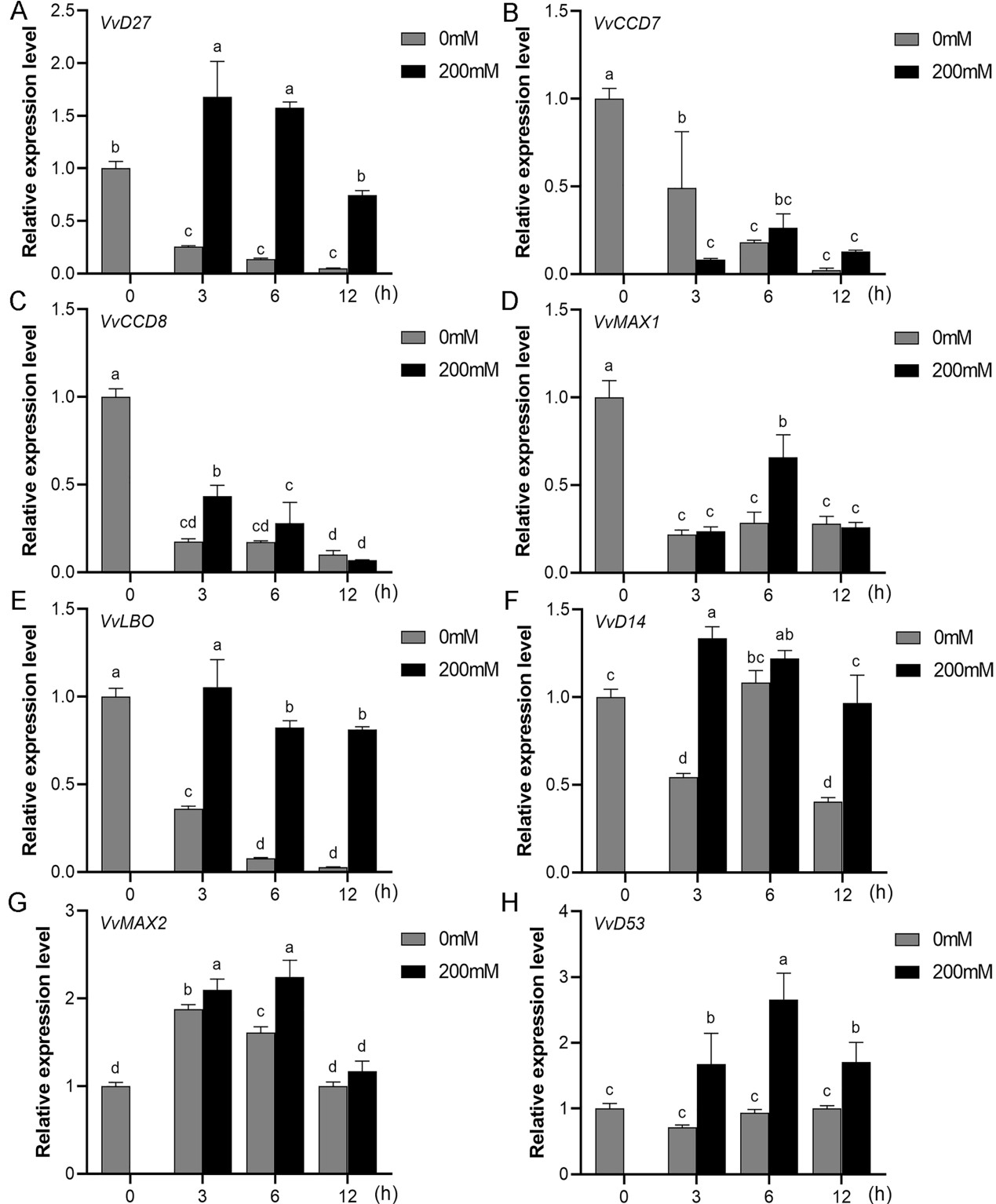

**Figure 10 The expression of grapevine SL biosynthetic and signaling genes in roots under salt stress.** (A) *VvD27*, (B) *VvCCD7*, (C) *VvCCD8*, (D) *VvMAX1*, (E) *VvLBO*, (F) *VvD14*, (G) *VvMAX2*, (H) *VvD53*. Grape seedlings were treated with 200 mM NaCl for 0, 3, 6 and 12 h, the roots of control and treated grapevine seedlings were selected for RT-qPCR analysis for the same treatment time. The mean expression value was calculated from three independent replicates.

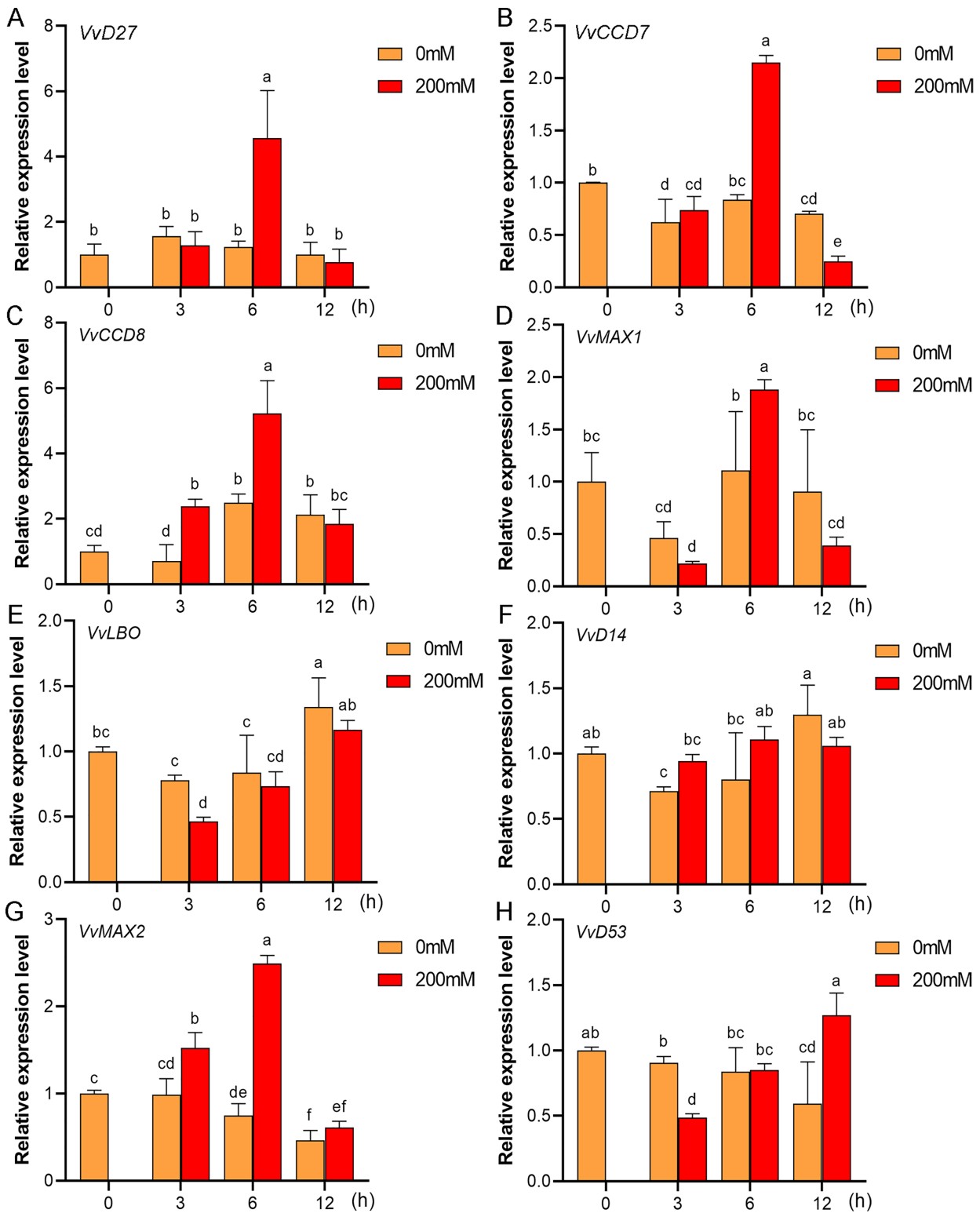

**Figure 11 The expression of grapevine SL biosynthetic and signaling genes in leaves under drought stress.** (A) *VvD27*, (B) *VvCCD7*, (C) *VvCCD8*, (D) *VvMAX1*, (E) *VvLBO*, (F) *VvD14*, (G) *VvMAX2*, (H) *VvD53*. Grape seedlings were treated with 200 mM mannitol for 0, 3, 6 and 12 h, the leaves of control and treated grapevine seedlings were selected for RT-qPCR analysis for the same treatment time. Vertical bars indicate the standard error of mean. Different lowercase letters indicate differences in gene expression according to the Duncan's multiple range tests with analysis of variance (ANOVA) ($p < 0.05$).

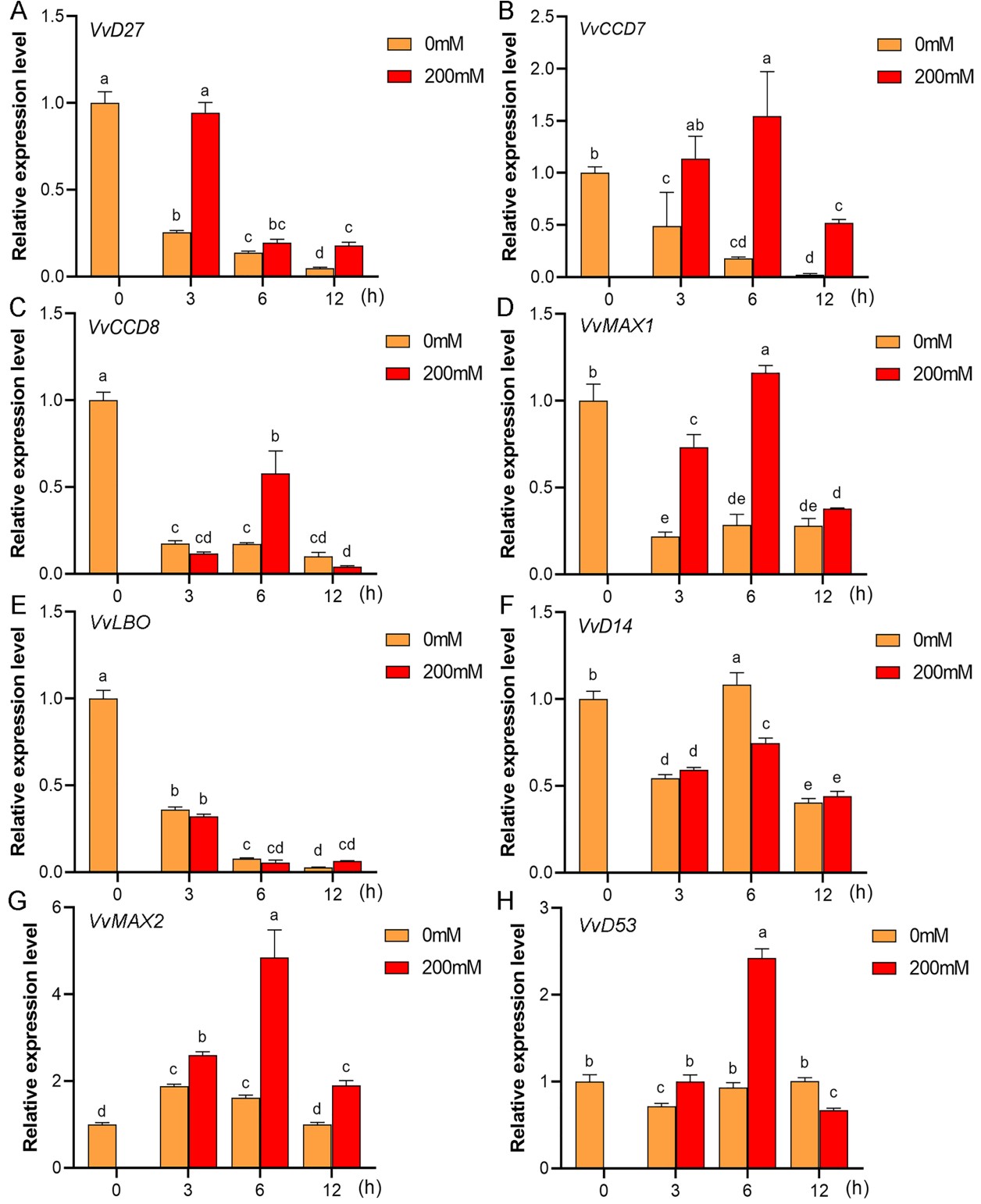

**Figure 12 The expression of grapevine SL biosynthetic and signaling genes in roots under drought stress.** (A) *VvD27*, (B) *VvCCD7*, (C) *VvCCD8*, (D) *VvMAX1*, (E) *VvLBO*, (F) *VvD14*, (G) *VvMAX2*, (H) *VvD53*. Grape seedlings were treated with 200 mM mannitol for 0, 3, 6 and 12 h, the roots of control and treated grapevine seedlings were selected for RT-qPCR analysis for the same treatment time. Vertical bars indicate the standard error of mean. Different lowercase letters indicate differences in gene expression according to the Duncan's multiple range tests with analysis of variance (ANOVA) ($p < 0.05$).

from some plants, such as rice, *Arabidopsis*, soybean, and straw berry (*Waters et al., 2017*; *Wu et al., 2019*; *Qiao et al., 2020*). However, the whole-grapevine SL biosynthetic and signaling genes have not been explored so far, and their potential functions in response to salt and drought stresses remain largely unclear. Therefore, in this study, the SL biosynthetic and signaling genes were identified in grapevine genome, and their possible roles in response to salt and drought were investigated.

Previous studies have reported that *Arabidopsis* have only one each of the *D27*, *CCD7*, *CCD8*, *MAX1*, *LBO*, *D14*, *D3* genes, and three *D53* genes (*AtSMXL6/7/8*), which involved in SL biosynthesis and signaling. In rice, there are one each of the *D27*, *CCD7*, *CCD8*, *LBO*, *D14*, *D3* genes, three *MAX1* genes (*OsMAX1a/b/c*), and two *D53* genes (*OsD53a/b*) (*Waters et al., 2017*). Woodland strawberry also contains only one *D27*, *CCD7*, *CCD8*, *LBO*, *D14*, and *D3* gene, respectively, but have two *MAX1* genes (*FveMAX1A* and *FveMAX1B*) and two *D53* genes (*FveD53A* and *FveMAX1B*) (*Wu et al., 2019*). The above results indicated that a number of gene duplications appear in the SL biosynthetic and signaling genes. Later research further confirmed this point, almost all soybean SL biosynthetic and signaling genes existed a pair of duplications, such as *GmD27a/b*, *GmCCD7a/b*, *GmCCD8a/b*, *GmMAX1a/b/c* (*Qiao et al., 2020*). However, in this investigation, only one each of *D27*, *CCD7*, *CCD8*, *MAX1* and *LBO* genes were identified as SL biosynthesis, and one each of *D14*, *MAX2* and *D53* genes were identified as related to SL signaling in grapevine genome (Table 1). These genes shared the almost similar identical motifs with their homologues in other plants, respectively, suggesting that SL related genes are conserved during evolution in different plant species. Further, the phylogenetic analysis explored that VvD27, VvCCD8, VvMAX1 and VvD53 has the highest homology to the same proteins of soybean, while VvCCD7, VvMAX2, VvLBO, VvD14 showed highest homology to AtCCD7, AtMAX2, MdD14, respectively. These results indicated that these proteins may have similar functions.

Mounting evidence revealed that SLs can positively regulate plant resistance to abiotic stresses (*Cardinale et al., 2018*; *Min et al., 2019*), and some SL biosynthetic and signaling genes have been found to participate in plant abiotic stress response either (*An et al., 2016*). In *Arabidopsis*, it was revealed that *AtMAX2*, *AtCCD7*, and *AtCCD8* can positively regulate plant response to salt and drought stresses (*Ha et al., 2014*). In soybean, *GmCCD7b*, *GmMAX1c*, *GmMAX2a/b* and *GmD53a/b* exhibited significantly higher expression levels under salt stress (*Qiao et al., 2020*). In apple, overexpression *MdMAX2* and *MdD14* genes could significantly enhance the plant tolerance to salt and drought (*An et al., 2016*; *Yang et al., 2019*). In the present study, the promoter sequences analysis indicated that the *cis*-acting elements related to various hormones and abiotic stresses widely existed in the promoter regions of these SL-related genes, such as ARE, MYB, MYC, ABRE, ERE and TC-rich repeats (Fig. 7), which suggested that SL biosynthetic and signaling genes may have potential ability to response the abiotic stress. Further, we found that the expression of eight grapevine SL related genes (*VvD27*, *VvCCD7*, *VvCCD8*, *VvMAX1*, *VvD14*, *VvMAX2*, *VvD53* and *VvLBO*) was upregulated both in roots and leaves under salt stress, and six grapevine SL related genes (*VvD27*, *VvCCD7*, *VvCCD8*, *VvMAX1*, *VvMAX2* and *VvD53*) showed a significantly increased expression both in roots and leaves under

drought stress. In contrast, *VvLBO* and *VvD14* displayed a downregulated expression under drought stress simulated by mannitol treatment. This finding illuminated that the grapevine SL biosynthetic and signaling genes may play important roles in responses to salt and drought stresses.

## CONCLUSIONS

In conclusion, five SL biosynthetic genes (one each of *D27*, *CCD7*, *CCD8*, *MAX1* and *LBO*) and three SL signaling genes (one *D14*, one *MAX2*, and one *D53*) were identified in grapevine genome. These SL biosynthetic and signaling genes are highly conserved during evolution among different plant species, and some *cis*-acting elements related to abiotic stress were prevalent in the promoter regions of these SL-related genes. Furthermore, the expression profile demonstrated that SL related genes may play important regulatory roles in response to salt and drought stresses response in grapevine. These findings provided valuable information for further investigation and functional analysis of grapevine SL biosynthetic and signaling genes in response to salt and drought stresses.

## ACKNOWLEDGEMENTS

We thank Prof. Yuanpeng Du of Shandong Agricultural University, China, for providing experimental materials and guidance. We greatly thank Tian Qiao, Lei Zhang and Changcheng Sui for experimental methods assistance.

### Funding

The present study was supported by the National Natural Science Foundation of China (31972358), the Natural Science Foundation of Shandong Province, China (ZR2018MC022), and the Shandong Provincial Key Research and Development Project (2019JZZY010727 and 2019GNC106147). The funders had no role in study design, data collection and analysis, decision to publish, or preparation of the manuscript.

### Grant Disclosures

The following grant information was disclosed by the authors:
National Natural Science Foundation of China: 31972358.
Natural Science Foundation of Shandong Province, China: ZR2018MC022.
Shandong Provincial Key Research and Development Project: 2019JZZY010727 and 2019GNC106147.

### Competing Interests

The authors declare that they have no competing interests.

### Author Contributions

- Yanyan Yu conceived and designed the experiments, performed the experiments, analyzed the data, prepared figures and/or tables, and approved the final draft.

- Jinghao Xu analyzed the data, prepared figures and/or tables, and approved the final draft.
- Chuanyin Wang analyzed the data, authored or reviewed drafts of the article, and approved the final draft.
- Yunning Pang performed the experiments, prepared figures and/or tables, and approved the final draft.
- Lijian Li performed the experiments, authored or reviewed drafts of the article, and approved the final draft.
- Xinjie Tang performed the experiments, prepared figures and/or tables, and approved the final draft.
- Bo Li analyzed the data, authored or reviewed drafts of the article, and approved the final draft.
- Qinghua Sun conceived and designed the experiments, analyzed the data, prepared figures and/or tables, and approved the final draft.

### Data Availability

    The raw data is available in the Supplemental Files.

### Supplemental Information

Supplemental information for this article can be found online at http://dx.doi.org/10.7717/peerj.13551#supplemental-information.

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
