# Peer review of "Genome-wide analysis of the strigolactone biosynthetic and signaling genes in grapevine and their response to salt and drought stresses"

_PeerJ, doi:10.7717/peerj.13551_

## Round 0.1 · original submission · Minor Revisions

Please address the reviewers' comments in your rebuttal letter and the manuscript.

Reviewer 1 ·

Basic reporting

The work adds more knowledge to the general understanding of grapevine strigolactone related genes under salt and drought stresses. I estimate that the paper can be recommended for publication in the PeerJ, after consideration of a few suggestions.

The authors need to consider doing a gene duplication events analysis of grapevine SL-related genes with Arabidopsis and rice, which could make the genome-wide analysis more informative.

Line 149-150:The TBtools software does not match the indexed literature (Krzywinski et al., 2009).

Experimental design

It is good designed.

Validity of the findings

The findings are interesting.

·

Basic reporting

The article is well written and thoroughly covered the literature. This a is a professional article with clear explanation of data.

Experimental design

The experimental design is well perceived.

Validity of the findings

The finding are novel and benefit to scientific community. The data is robust and statistically sound.

Additional comments

I have following comments to improve the quality of the manuscript:

1) Can the authors comment on why vvD27 showed increased expression at high salt concentration even after 12hrs (Fig 7), in contrast to other genes expression profiles (All other genes showed elevated expression until 6hrs and reduced expression after 6 hrs).

2) The authors annotated motifs of protein corresponding to a gene as motif1. motif2, motif3...etc. Instead, if the authors annotate these motifs based on their unique chemical or biological function, it would be more informative.

3) The authors demonstrated through phylogenetic analysis that The results showed that VvD27, VvCCD8 and VvMAX1 share higher homology with corresponding proteins in soybean. To better support the data, the authors should also provide expression profiles of soybean genes and compare with that of grapevine SL biosynthetic proteins (as shown figure 6).

4) It is not clear how the authors generated SL-related protein sequences from the genome data obtained from Grapevine Genome Browser? Also, did the authors perform BLAST analysis on nucleotide sequence or protein sequence?

Minor comments:

1) Line 90 need to be corrected as - "Recent studies have revealed that some SL biosynthetic and signaling genes play critical roles in abiotic stresses response."

2) Although the authors briefly described in line 63-66, it would be beneficial if the authors prepare an image depicting the Carotenoid pathway.

3) Line 63: Instead of using the word "Firstly", it would be better to use "Initially"

4) Line 47-49 is a broad statement, which require numerous citations. Please review the literature thoroughly and cite all relevant citations. For eg: https://www.pnas.org/cgi/doi/10.1073/pnas.1817233116

---

## Round 0.2 · accepted · Accept

Thank you very much for addressing the reviewers' comments and addressing the remaining comments.

·

Basic reporting

no comment

Experimental design

no comment

Validity of the findings

no comment

Additional comments

The authors have responded to all of my concerns in an efficient manner. So, I don't have any further comments.